# Graphene Oxide, Carbon Nanotubes, and Polyelectrolytes-Based Impedanciometric E-Tongue for Estrogen Detection in Complex Matrices

**DOI:** 10.3390/s24020481

**Published:** 2024-01-12

**Authors:** Tiago Reis, Maria Helena Fino, Maria Raposo

**Affiliations:** 1Laboratory of Instrumentation, Biomedical Engineering and Radiation Physics (LIBPhys-UNL), Department of Physics, NOVA School of Science and Technology, Universidade NOVA de Lisboa, 2829-516 Caparica, Portugal; tff.reis@campus.fct.unl.pt; 2Centre for Technology and Systems (LASI-CTS), UNINOVA, Department of Electrotechnical and Computer Engineering, NOVA School of Science and Technology, Universidade NOVA de Lisboa, 2829-516 Caparica, Portugal; hfino@fct.unl.pt

**Keywords:** e-tongue, graphene oxide, polyelectrolytes, layer-by-layer films, impedance spectroscopy, emerging pollutants, estrogen, 17β-estradiol, complex matrices, principal components analysis

## Abstract

Currently, it is necessary to maintain the quality of aquifers and water bodies, which means the need for sensors that detect molecules as emerging pollutants (EPs) at low concentrations in aqueous complex solutions. In this work, an electronic tongue (e-tongue) prototype was developed to detect 17β-estradiol in tap water. To achieve such a prototype, an array of sensors was prepared. Each sensor consists of a solid support with interdigitated electrodes without or with thin films prepared with graphene oxide, nanotubes, and other polyelectrolytes molecules adsorbed on them. To collect data from each sensor, impedance spectroscopy was used to analyze the electrical characteristics of samples of estrogen solutions with different concentrations. To analyze the collected data from the sensors, principal components analysis (PCA) method was used to create a three-dimensional plane using the calculated principal components, namely PC1 and PC2, and the estrogen concentration values. Then, damped least squares (DLS) was used to find the optimal values for the hyperplane calibration, as the sensitivity of this e-tongue was not represented by a straight line but by a surface. For the collected data, from nanotubes and graphene oxide sensors, a calibration curve for concentration given by the 10^PC1×0.492−PC2×0.14–14.5^ surface was achieved. This e-tongue presented a detection limit of 10^−16^ M of 17β-estradiol in tap water.

## 1. Introduction

Nowadays, personal care and pharmaceutical products (PPCP), also referred to as emerging pollutants (EPs), are acknowledged as an environmental threat since they are being accumulated in aquifers and water bodies [1,2,3,4,5,6,7,8,9,10,11,12,13,14]. Although many water treatment facilities have been built for their purification, new EPs are still being discovered, and there is a lack of sensors that can properly monitor them with a reasonable maintenance cost, meaning these facilities are unfit and lack the means to fulfill their purpose in its integrity [15].

The only feasible solution to this problem is to improve the facilities with new sensors capable of accurately detecting these new EPs. The new sensors should be small, capable of real-time analysis, and inexpensive. Additionally, for a sensor to be fit for this type of application, it also needs to be capable of identifying the concentration of a specific molecule present in a complex solution. This is because water is composed of, in addition to water molecules, many different salts and substances [1,15,16,17].

Estrogen is an endocrine-disrupting chemical (EDC); it has been already confirmed that these molecules cause harm to humans in the form of “cancer, developmental problems, diabetes, and possibly also obesity and the metabolic syndrome” [17], among other possibilities. It has also been proven that estrogen is present in riverbank infiltrations [15] and sewage [16], confirming the need for a sensor capable of detecting it [18,19].

The detection of estrogen has been a relevant topic for over two decades now. But, since 1996, the yeast estrogen screen method (YES) that employs chemical sensors has been successful in detecting low concentrations of estrogen [20]. Later, in 2018, the Arxula yeast estrogen screen method, an improvement on the classic YES method, was validated, and it proves relevant still [21]. This method can detect the concentration of estrogen-like substances by making them react with estrogen reactors (ERs) and measuring the result. However, it requires sampling and long periods of incubation in a lab. This increases its cost and requires transportation of the samples without providing real-time analysis. Additionally, the method of gas chromatography–mass spectrometry was used to determine the concentration of EE2 in the central south coast of Chile [15]; in this study, values between 4.18 and 48.14 ng/g dry weight were obtained.

To overcome the above-mentioned limitations, e-tongue systems are proposed to detect estrogen, as this type of systems has been able to detect low concentrations of other substances in complex solutions [22,23,24,25,26,27,28]. An e-tongue system consists of an array of sensors; an electrical system for the measurement of electrical characteristics of the aqueous solutions, for example, a potentiometer or an impedance analyzer; and a computer. With this arrangement, electric signals can be sent to the sensors while monitoring their electrical response when immersed in the aqueous samples [26,27,29,30,31,32,33,34,35]. These studies demonstrated that the electric behavior of electrical sensors can be appraised from the impedance and conductivity properties of the sensors and of the aqueous solutions analyzed [27,28]. This implies that, if the sensor is an appropriate transductor to a substance, changes to the values of the impedance and conductivity are correlated with the presence of the substance to be detected. Recently, Paulo Zagalo et al. [19] demonstrated that the thin films polyethyleneimine (PEI) and poly (sodium 4-styrene sulfonate) (PSS) can be used to detect low concentrations of 17α-ethinylestradiol. Following these results, the main objective of this work was to create an e-tongue system containing an array of different sensors to detect 17β-estradiol in aqueous complex matrices such as tap water. Each sensor consists of interdigitated electrodes deposited on a solid support covered or not by sensorial layers capable of receptivity to the presence of a predefined substance. For that, the sensorial layers were prepared with the polyelectrolyte’s poly (allylamine hydrochloride) (PAH), PEI, poly{1-[4-(3-carboxy-4-hydroxyphenylazo) benzene sulfonamido]-1,2-ethanediyl, sodium salt} (PAZO), multi-walled carbon nanotubes (mwcnt), and graphene oxide (GO). Different combinations of polyelectrolytes were tested regarding their ability to detect the estrogen concentration in tap water. The data acquired from the multiple sensorial devices allowed a multivariate analysis. To extract as much information as possible from each sensor, usually, the electrical measurements consist of the impedance spectra. The result is a series of “coma-separated values” (csv) files, where each file represents a concentration measured with a sensorial device. These data were analyzed by principal components analysis (PCA) and damped least squares (DLS) methods [26,27,29,30,31]. The developed e-tongue proved able to detect very low concentrations of estrogen molecules up to 10^−16^ M.

## 2. Materials and Methods

### 2.1. Sensor Array

A sensor can produce an output signal based on its characteristics that should be sensitive to the environment around the sensor [36]. As the aim was to develop an e-tongue to detect 17β-estradiol, we firstly focused on developing sensors that react to the presence of this estrogen.

To produce a sensor capable of reacting to different concentrations of a molecule, besides the electrodes, a sensorial layer is used to increase the sensitivity and reproducibility. The sensorial layer makes it possible for the sensor impedance to change depending on the number of certain molecules present in its surroundings. The sensorial layer consists of thin films that allow it to react discriminately or not toward a target substance.

Nanostructured sensorial layers based on layer-by-layer (LbL) thin films of polyelectrolytes were deposited on the sensorial area of ceramic substrates previously coated with gold interdigitated electrodes devices acquired from Metrohm DropSens (Oviedo, Asturias, Spain) [24]. The polyelectrolytes used were PAH, PAZO, PEI, poly(sodium 4-styrenesulfonate) (PSS), GO, and mwcnt, and the respective chemical structures are presented in Figure 1a–e. These polyelectrolytes are common in sensor development and used in the references [19,22,23,24,25,37,38]. The choice of these polyelectrolytes was based on the fact that the preparation of these films is already well established, and the same film can detect different molecules, as has already been demonstrated.

For the preparation of the sensorial layers of controlled thickness, the inexpensive and precise LbL method was used [39]. This method is based in adsorption of polyelectrolytes’ layers of opposite electrical charges at solid liquid interface. The procedure for preparation of LbL films by applying the LbL method is described in [39] and consists of the immersion of the solid support with interdigitated electrodes in the polycationic solution for 60 s, washing the adsorbed layer in ultra-pure water, and drying it with flux of nitrogen; next is the immersion of the solid support with the layer adsorbed in the polyanionic solution for 60 s followed by washing the solid support with the bilayer al-ready adsorbed in ultra-pure water and then drying it again with a flux of nitrogen. This process is repeated as many times as the desired number of the different layers considered. The combination of one layer from each of the two different polyelectrolytes is called a bilayer. The sensitivity of the film will be dependent on the compounds used, the number of bilayers, and the thickness of each layer. The thickness is dependent on immersion time and the monomeric concentration of the polyelectrolyte solution. In the present case, solutions of PAH, PAZO, PEI, PSS, mwcnt, and GO with a monomeric concentration of 10^−3^ M were prepared.

### 2.2. Estrogen Solutions

To prepare tap water samples with diverse and precise concentrations of estrogen, a solution with a high concentration was prepared and diluted. The base solution has a concentration of 10^−5^ mol/L and was prepared by weighing 0.27 mg of 17β-estradiol and mixing it in 100 mL of methanol (MH_3_OH). Whenever a set of concentrations was to be prepared, first, 1 mL of the base solution was mixed with 9 mL of tap water and 1 mL of methanol; then, this dilution was repeated until all the desired concentrations were prepared. To maintain a constant concentration of 10% methanol, only 0.9 mL of methanol was added in the sequential dilutions. The tap water used in all the solutions was obtained from the tap and immediately used to prepare the solutions.

### 2.3. Impedance Measurements

The electrical properties of the developed sensor devices were analyzed by impedance spectroscopy. The impedance analyzer used in this work was the Solartron 1260 Impedance/Gain-Phase Analyzer, coupled to a 1296A Dielectric Interface module. The analysis parameters are defined in the SMaRT Impedance software (version 3.3.1).

For reasons explored in Section 2.4, it is important to constantly maintain the level of immersion and temperature while measuring. To maintain a constant level of immersion of the electrode throughout the measures, the electrode was immersed in 2 mL contained in a 5 mL beaker until it reached the beaker’s floor. To constantly maintain the effects from the temperature, the measurements were taken right after the solution was removed from the fridge, where it was stored at 4 °C, and the solution was stored in it again right after pipetting. This way, even if the second and third loops suffered some variation due to the temperature variation, as the solution stayed exposed to room temperature for some minutes while the measurements were being taken, it was neglectable compared to the variation resulting from the degradation of the film and constant throughout the different films and different concentrations.

### 2.4. Effect of the Electrode’s Area Immersed in the Sample Solutions

To confirm the effect of the immersion and emersion of the interdigitated electrode on the measurements, an experiment was conducted by measuring the impedance spectra by immersing the interdigitated electrodes in different positions, namely in four possible positions, as illustrated in Figure 2. It was immersed from one position to the other, until it was fully immersed at position p4, capturing the effect of those positions when the upper (not immersed) part of the electrode was dry. After that, the positions were measured again while lifting the electrode, and thus, it was possible to see the effect of leaving the upper part wet when taking a new measurement.

The results are presented in Figure 2, where the measures taken while immersed the electrode are identified with a _1, and the measures taken when emerged are identified with _2.

Although for each measurement taken, three consecutive measurements were taken, to improve the readability of the result, only the first one is presented. This does not compromise the accuracy of the results because the margin of error between these three measurements of the same position was negligible compared to the difference between the positions.

To maintain a consistent level of immersion of the electrode throughout the measures, the electrode was immersed in 2 mL contained in a 5 mL beaker until it reached the beaker’s floor. To constantly maintain the effects from the temperature, the measurements were taken right after the solution was removed from the fridge, where it was stored at 4 °C, and the solution was stored in it again right after pipetting.

### 2.5. Data Treatment

The principal component analysis (PCA) is a statistical algorithm used to extract important information from data represented in a matrix containing observations described by multiple inter-correlated quantitative variables. This algorithm starts by autoscaling all the values in each column, resulting in zeros representing the average of the column. Then, it calculates the optimal weights by solving the eigenvector problem for the covariance matrix.

By trying to identify the same concentration with multiple sensors simultaneously, a redundancy in the dimensions of the dataset is created. To solve this, e-tongue training can be optimized by reducing its dimension. PCA was the main algorithm used for this purpose.

PCA makes it possible to convert the impedance values of each sensor into a variable that can be calibrated to deduce/predict the concentration. This variable will be the result of a principal component (PC) or a mix between them. PC1 is the vector of weights that best preserves the distance between the values in the dataset. PC2 is perpendicular to PC1 and maximizes the distances within this condition. PC3 is the vector for the plain described between PC1 and PC2; if the dataset has fewer than three dimensions, this will be a null vector. Once the dimension of the dataset has been reduced, if a pattern emerges from the PCs, it is possible to predict the concentration by calibrating the sensor.

The maximization is obtained by resolving an eigenvalue problem because the covariance matrix has eigenvectors that point in the direction that maximizes the variance. The covariance matrix is an *n* × *n* matrix, where *n* is the number of variables in the dataset, and it holds the variance of each variable along the diagonal and the covariance of each variable with the others through the variable’s respective column. An eigenvector is a matrix whose direction is preserved when multiplied by its matrix, and a scale factor between the result and the original is observed; this scale is called the eigenvalue. To solve this type of problem, first, the eigenvalues are determined by finding the determinant at which the covariance matrix minus the identity matrix equals zero (1). Then, by applying the definition of the eigenvector mentioned above, we can solve the system of equations that makes the product between the covariance matrix and the eigenvector match the product between one of the possible eigenvalues and the eigenvector (2). Finally, because the solution gives only the direction of the eigenvector, we define it with absolute values that make it a unit vector. This algorithm is explained in further detail in references [40,41,42].(1)$$\det\left(covariance_{Matrix}-\lambda \times I\right)=0,$$
(2)$$covariance_{Matrix}\times eigen_{Vector}=\;eigen_{Value}\times eigen_{Vector},$$

Using classes in the discrete domain to represent the different concentrations creates a discretization error, which occurs when a function of a continuous variable is represented in the computer by a finite number of evaluations [43]. To avoid this error, the sensor’s calibration will be the result of a fitting algorithm, i.e., a continuous function that approximates the training set. Curve fitting is based on regression analysis, and it can be used by solving the least squares problem. The least squares problem is the minimization of the sum from all the squared errors. The more common methods for solving this type of problem are the damped least squares (DLS) and the Gauss–Newton algorithm (GNA). DLS and GNA are similar, and the main difference between them is that DLS uses the gradient descent method. The gradient descent method serves to find local minimums and helps DLS be more robust than GNA. However, this also means DLS is slower than GNA. In the context of machine learning, it is common to prioritize accuracy over speed, as the time it takes to calibrate the module and the time the module takes to produce results are independent.

DLS can also be called the Levenberg–Marquardt algorithm (LMA or just LM), and it is an iterative procedure. It takes an initial guess of the parameters as a seed, and in cases with multiple minimums, it can only convert to the global minimal error if the initial seed is good. There already exist open-source routines that solve least squares problems using the DLS, making it easily accessible.

In this work, DLS was used through the Python function *curve_fit* from the *SciPy.Optimize* library to calculate all the trendlines presented in the results as well as the calibration surface used with the PCA.

## 3. Results

### 3.1. Effect of Repeated Measurements

The preliminary results demonstrated that the sensorial layer degrades with the measuring time since the film with the passage of electrical current reacts with the solution where it is immersed. Passing an electric current through the sensorial layer triggers chemical reactions, damaging the film’s molecules. While the sensor is immersed in an aqueous solution, the thin film can also desorb over time. When measuring the impedance spectra, one loop takes approximately 1 min and 20 s. As such, it is expected that most of the damage done to the sensorial layer comes from passing electric current through the thin films.

In this section, first, an initial analysis of the degradation with the (PAH⁄PAZO)_5_ film is presented. Then, a complete analysis of the degradation of the films (PAH⁄PAZO)_5_, (PEI⁄GO)_5_, (PEI⁄mwcnt)_5_, and (PEI⁄PSS)_5_ is presented.

To observe and characterize the effect of the degradation on the sensor presented in the results from the sensor with a (PAH⁄PAZO)_5_ film, two films of (PAH⁄PAZO)_5_ were prepared and immersed in the estrogen concentrations of 0 M and 10^−6^ M. Then, each was measured over two sets of 15 loops; every set of loops lasted 20 min. The results of the 60 measurements are presented in Figure 3. The points from 10^−6^ M and 0 M are represented by orange and blue, respectively. The points to which more loops correspond are represented with a lighter color.

To better visualize the data, certain frequencies are fixed, and the results are plotted over several loops. The chosen frequencies are the ones with either the highest or the lowest variance between the loops. However, the frequency where either the highest or the lowest variance in the degradation is observed can be different depending on the concentration, but to compare them, they must be plotted at the same frequency. As a compromise, they are fixed at the frequency where either the highest or the lowest variance in the degradation is observed between the two concentrations. This results in four graphs: two for the impedance and two for the phase, as illustrated in Figure 4.

All the deviations in these measurements can be attributed to either degradation or temperature change. As the samples were stored in a fridge and measured before reaching room temperature and were exposed to a higher room temperature for over 40 min, the rise in their temperature is not negligible. However, if the temperature was the only effect to cause deviation, both samples should present similar tendencies, but that is not the case. The conclusion reached is that the sensor suffers more degradation with higher concentrations.

As such, the experiment was repeated with the solutions at room temperature, removing the temperature variable, to confirm the effect of the degradation by the number of loops. The results are illustrated in Figure 5a.

These results reveal that most of the variance in the previous analysis was the result of the temperature change, confirming that both the temperature and the number of measurements taken influence the results. The same analysis of the degradation was repeated for films of (PEI⁄GO)_5_, (PEI⁄mwcnt)_5_, and (PEI⁄PSS)_5_, as illustrated in Figure 5b–d, respectively.

These results show that (PEI⁄PSS)_5_ and (PEI⁄GO)_5_ suffer great degradation for lower frequencies, where there is the best resolution; however, only (PEI⁄PSS)_5_ shows insignificant discrimination between the extreme concentrations for the first loop. As such, this analysis suggests that the film (PEI⁄PSS)_5_ is incapable of discerning the concentration of estrogen in tap water.

### 3.2. Results with Individual Films

To study each film through different concentrations of estrogen in tap water, twelve (PEI⁄GO)_5_, (PEI⁄mwcnt)_5_, and (PAH⁄PAZO)_5_ LbL films were prepared on solid supports with interdigitated electrodes. They were used to sample four concentrations of 10^−10^, 10^−13^, 10^−16^, and 0 mol/L three times each. The concentration of 0 mol/L is considered at a concentration of 10^−19^ to fit within the plot of the trend line, but this presumption is not a gross error, as the sample is tap water, and it is also expected that it would already contain estrogen, possibly in a concentration higher than the 10^−19^ considered. The results extracted from these sensors as well as a control interdigitated electrode, without any films, are presented in Figure 6.

For better readability, Figure 7 and Table 1 present the impedance shown in Figure 6 but fixated at the frequencies where the largest variations can be observed. The point with the concentration of 10^−19^ M was not considered when plotting the trend line, as it represents the control with an unknown concentration. As such, there is a big discrepancy between where the tendency predicts the point to be and where it is.

With these results, some conclusions can be drawn for each film. In the film (PEI/GO), the phase shows significant discrimination between the concentrations. For the film (PEI/mwcnt), it can be concluded that both the amplitude and the phase of the impedance were sensitive to the variation in the concentration for samples 1 and 3. Among the samples collected with (PAH/PAZO), sample 3 showed a different behavior from samples 1 and 2 in both the impedance and the phase, which indicates that its point for 10^−10^ M concentration could be an outlier. As such, for the (PAH/PAZO) film, the phase showed a clear tendency along the concentration. However, the amplitude did not show significant discrimination between the different concentrations relative to the margins of error. Additionally, in all films, the effects of the degradation could only be observed in the loops, as each point had an unused film. And, as expected, the trend lines between the loops are almost parallel to each other since the effect from the degradation is considerably consistent within a small number of loops.

### 3.3. Initial PCA Results

To test if using information from multiple frequencies can improve the performance of the features from these films, an initial PCA was used. The results were four plots for each feature monitored: the first plot was where all samples were individually normalized and their optimal weights used to obtain the PCs; the second plot was when all samples used the average and standard deviation from sample 1 to normalize themselves, but then their weights were calculated individually; the third plot was the reverse from the second, as each sample was normalized individually, but the weights from sample 1 were used; and finally, the last plot contained all samples and used the normalization and weights from sample 1.

The last plot where all samples used the normalization and weights from sample 1 was the most relevant, as it was used to test how the sensor would react to new data since a single new line of data cannot be normalized or used for PCA. To calculate the normalization parameters for the samples aside from sample 1, the three loops from sample 1 were normalized individually along the concentrations, and their means and standard deviations were averaged. To calculate the weights to use on the samples other than sample 1, PCA was used for each concentration of each loop in the first sample, and the three sets of weights calculated were averaged.

The best results from this analysis for the films (PEI/mwcnt) and (PAH/PAZO) was the real part of the impedance, and for the (PEI/GO) film, it was the loss tangent. The plots are illustrated in Figure 8.

With these results, it can be concluded that all films were on average sensitive to a variation in the concentration of estrogen. For the film (PEI/GO), sample 1 had a negative correlation contrary to the other samples; however, this could be attributed to the high margins of error. The film (PEI/mwcnt) had the best results without sample 2, but even with it, both the amplitude and the phase were sensitive, although the amplitude had relatively large margins of error. The (PAH/PAZO) film generally had the worst results, but in this analysis of the real part of the impedance, it had some sensitivity, although it still had the largest margins of error. However, because in this analysis, only the data from sample 1 were used to calculate the normalization and weights, the margins of error can either increase or decrease.

To obtain a better understanding of how the samples interact together, another PCA was used. The only difference made to the data structure was that the control samples were used in the columns. As there was one control sample in this case, the number of columns doubled. In this analysis, the focus was not on the correlation between PC1 and concentrations but on understanding the normalized dataset to be fed into the final PCA. As such, the plots show the normalized values along the columns as well as the weights given to them. For better readability, the control and the film are separated, and the columns are represented by their values in the frequency. The concentrations are differentiated by color, as indicated in the legend. Figure 9 illustrates the loss tangent for the three films. Within these plots, the best results come from the control of the nanotubes, as the different concentrations are separated and organized in a coherent order.

### 3.4. Training with PCA

To test the system’s capacity to discriminate between different concentrations of estrogen, PCA was used with the data from the fin films that were tested. PCA was used to condense the relevant features into a two-dimensional plane that allowed the visualization of the dataset. When reducing the dimensionality of a dataset, information is inevitably lost, but PCA minimizes this loss. The result from the PCA with the three films is illustrated in Figure 10a; however, as the (PAH/PAZO) film did not show aptitude to discriminate the different concentrations with the phase, Figure 10b illustrates a PCA without this film. Without the (PAH/PAZO) film, the PC1 no longer overlaps the points between concentrations of 10^−16^ mol/L and 10^−13^ mol/L.

The heatmap shows the predicted decadic logarithm for the concentration and represents the calibration plain. The equation to determine the concentration for any point is C = 10^PC_1×s_1+PC_2×s_2+constant^, where s_1 and s_2 are the sensitivities for PC1 and PC2, respectively.

The table used for PCA consists of 36 lines (3 loops × 4 concentrations × 3 samples) and 125 columns (name + 2 environments × 1 domain × 31 frequencies × 2(film + no film)). The lines correspond to the points, and each point is a loop of the impedance spectra from a concentration in one of the samples with the film. The first column has the names of all points, and the rest have the relevant features from each film; in this case, they use all 31 frequency values from the loss tangent as well as the values from the control sample. This means that the different loops and samples generate more points, while their PC values are determined by the loss tangent from the two films, i.e., (PAH⁄mwcnt) and (PEI⁄GO), as well as the two respective control samples. For the lines, only three samples, considered as the control samples, are used in the columns to serve as a reference parameter; this implies that the information from the control sample is repeated between the lines of non-control samples. The names used to represent the concentrations 10^−10^, 10^−13^, 10^−16^, and 0 (mol/L) are M10, M13, M16, and M0, respectively, and M0 was considered 10^−19^ for calibration purposes.

To understand how the PCA reached its results in Figure 10b, the weights attributed throughout its columns need to be illustrated. For better readability, the columns were separated and organized by experiment, feature, and either film or respective control. Figure 11 illustrates these results.

In Figure 11, the left data axis is used for the points that represent the normalized dataset used in the PCA, and the right axis is used for the weights attributed by the PCA. The different colors represent the same concentrations they did for Figure 10a,b; different symbols in the data represent different samples; and the transparency increases with the number of loops. The weights are scaled to fit between –1 and 1, and the blue and red weights are used to calculate PC1 and PC2, respectively.

The weights have higher absolute values in regions where the different concentrations are well differentiated. The graphs related to the control appear to have fewer points, but they simply overlap due to the repetition mentioned above. For the films, PEI/GO shows more consistent variances for each concentration compared to PEI/mwcnt, but unexpectedly, the biggest contribution was the values from the control sample associated with the nanotubes.

Overall, the heatmap illustrated in Figure 10b suggests that the system can separate the different concentrations with precision, as was intended. This is further supported by the patterns illustrated in Figure 11.

However, in this training, each film was only sampled once in tap water, and different samples of tap water contained different compositions. The effect caused by the difference in compositions was not tested, and as such, there is no certainty that this calibration is valid. Additionally, each film was tested only once, meaning this training also has no statistical relevance. As there is only one test from each film, it is impossible to test the trained data and predict the system’s performance.

Further testing of the effects from different samples of tap water would allow discerning whether the sensors of the e-tongue should be sampled in either the same or different samples of tap water. By collecting more samples for training and testing, it could also be concluded if more sensors need to be added to the e-tongue.

## 4. Conclusions

In this work, an e-tongue was developed to detect low concentrations of estrogen in aqueous complex solutions. Estrogen is a dangerous EP in the EDC group, and it needs to be detected in small concentrations. To produce an accessible sensor that can detect these molecules in real time, this work approaches the problem by using interdigitated electrodes covered or not with thin films. Interdigitated electrodes deposited on a solid support are capable of real-time analysis and are inexpensive. Additionally, when arranged as an array of sensorial devices in an e-tongue system, they can detect low concentrations of molecules in complex solutions, and thus, the devised e-tongue system was presented.

As such, the electric tongue system can accurately predict the value of estrogen present in any given solution. To achieve this, impedance spectroscopy was used, and the information through the frequencies was compiled. Determining the weights of each frequency for each parameter for each electrode was carried out with PCA. The calibration was completed with DLS.

The results based on a sensorial device with a film of (PEI/mwcnt)_5_ showed that the phase and loss tangent present the best reproducibility. Additionally, no use was found for the Nyquist plot in the detections of estrogen.

After collecting the samples, the PCA was used to calculate the weights for PC1 and PC2. Then, PC1, PC2, and −log10(concentration) were plotted, and the calibration concentration equal to 10^PC1×0.492−PC2×0.14–14.5^ was reached using (PEI/mwcnt)_5_ and (PEI/GO)_5_ thin-films data.

For future prospects, at least four more samples should be collected using the thin films to prepare a significant dataset. Using these new samples, three could be used for calibration and one for testing. As such, four combinations could be tested this way, and either the calibration with the best result or an overall average could be used. Additionally, by collecting more samples regarding degradation, drift compensation algorithms could be used to extend the sensors’ life cycle past single use [44,45].

## Figures and Tables

**Figure 1 sensors-24-00481-f001:**
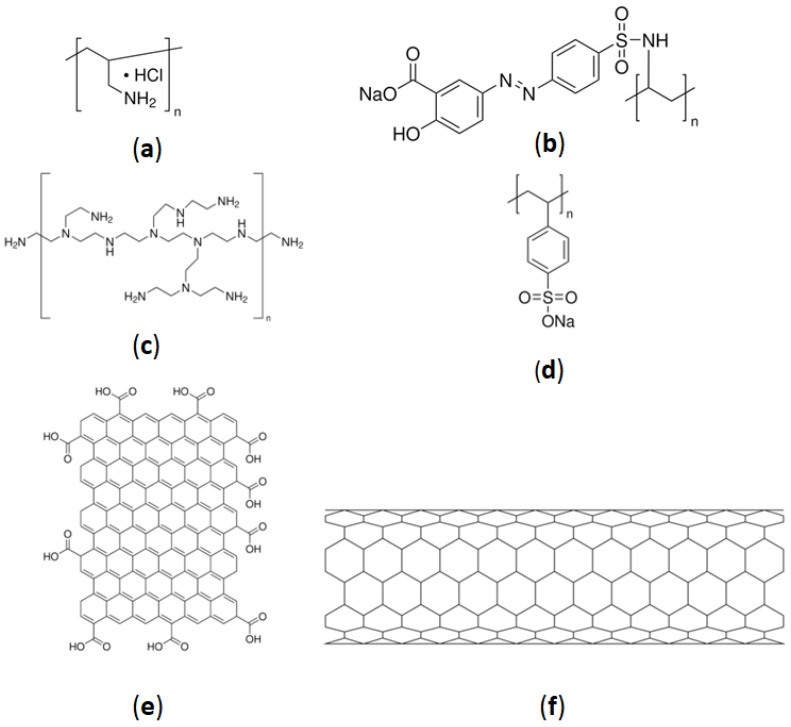
Chemical structures: (**a**) PAH; (**b**) PAZO; (**c**) PEI; (**d**) PSS; (**e**) GO; (**f**) mwcnt.

**Figure 2 sensors-24-00481-f002:**
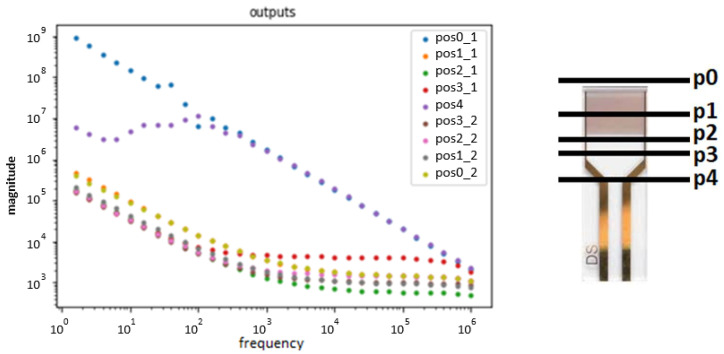
Absolute impedance over frequency for every position.

**Figure 3 sensors-24-00481-f003:**
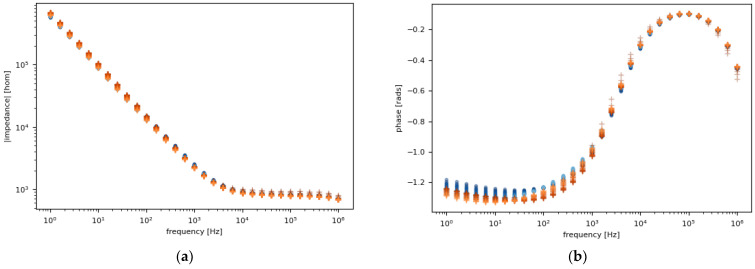
(**a**) Impedance magnitude; (**b**) phase spectra measured continuously (15 loops) to analyze the (PAH⁄PAZO)_5_ sensor’s degradation. The points from 10^−6^ M and 0 M are represented by orange and blue, respectively. The points corresponding to more loops are represented with a lighter color.

**Figure 4 sensors-24-00481-f004:**
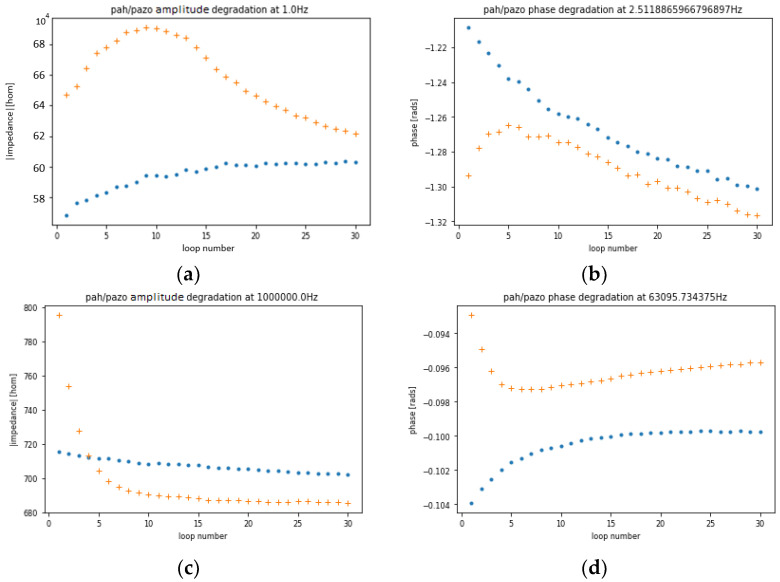
(**a**) Impedance magnitude at1 Hz; (**b**) phase at 2.5 Hz; (**c**) impedance magnitude at 1 MHz; (**d**) phase at 63 kHz as a function of number of loops in measurements of (PAH⁄PAZO)_5_ film deposited on the sensor device. Blue and orange points correspond to estrogen concentrations of 0 M and 10^−6^ M, respectively.

**Figure 5 sensors-24-00481-f005:**
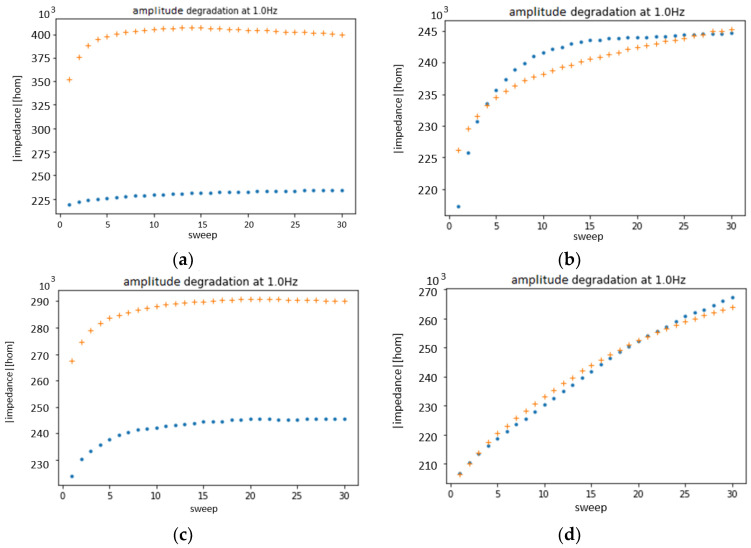
Impedance magnitude at 1 Hz as a function of the number of loops in measurements of (**a**) (PAH⁄PAZO)_5_, (**b**) (PEI⁄GO)_5_, (**c**) (PEI⁄mwcnt)_5_, and (**d**) (PEI⁄PSS)_5_ film deposited on the sensor device. Blue and orange points correspond to 0 and 10^−6^ mol/L, respectively.

**Figure 6 sensors-24-00481-f006:**
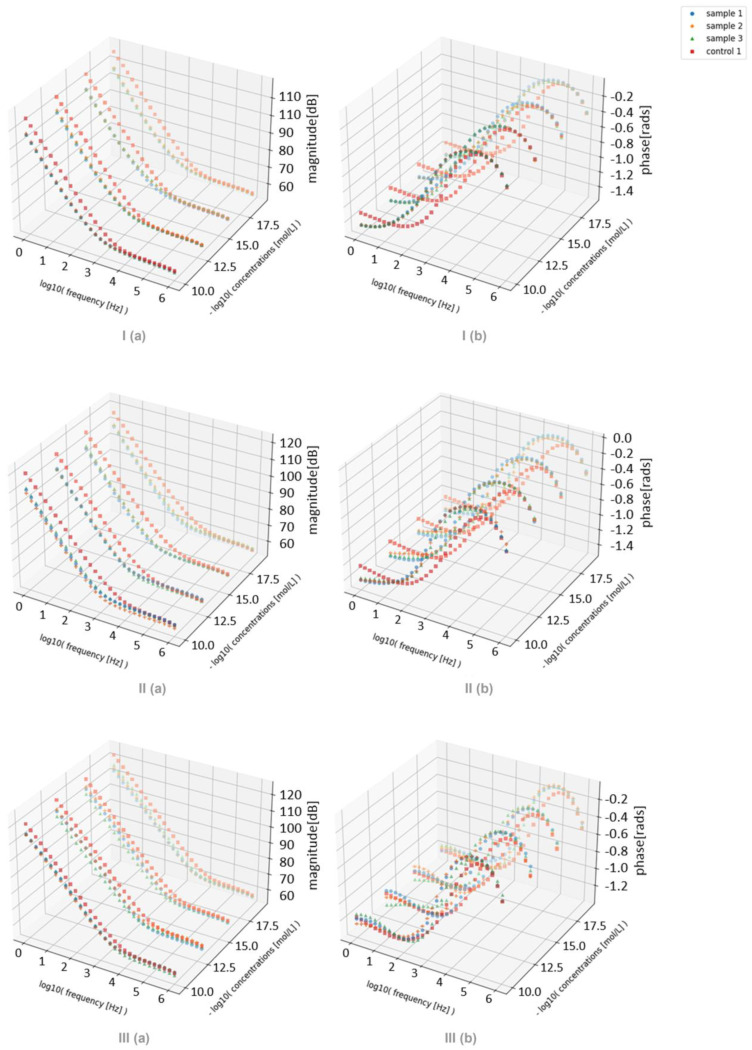
(**a**) Impedance magnitude spectra; (**b**) phase spectra of sensor device with (I) (PEI/GO)_5_, (II) (PEI/mwcnt)_5_, and (III) (PAH⁄PAZO)_5_ film immersed in aqueous solutions at different estrogen concentrations.

**Figure 7 sensors-24-00481-f007:**
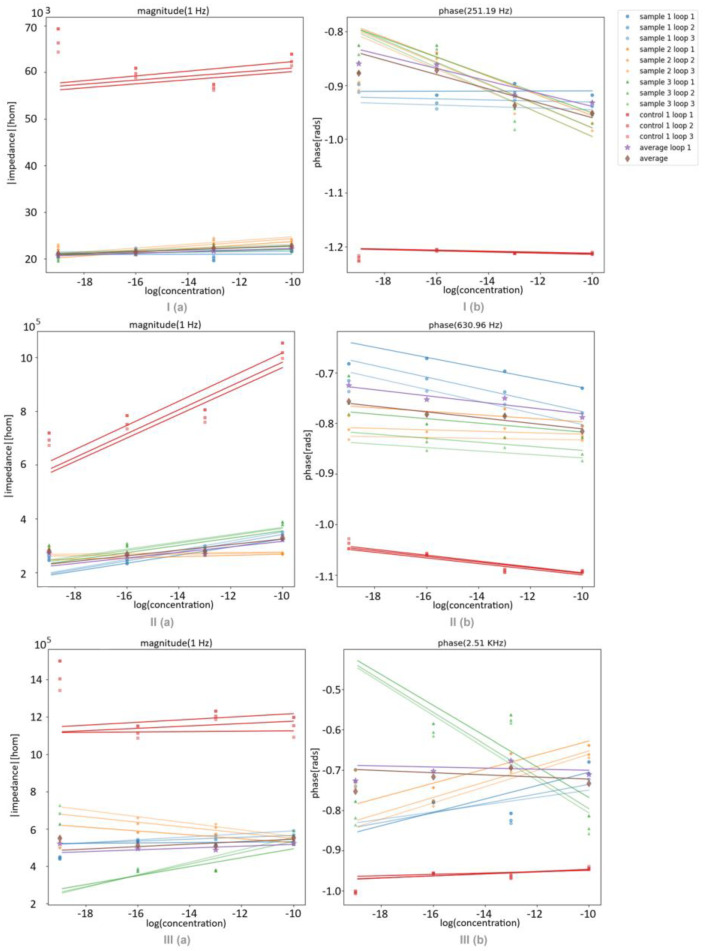
(**a**) Impedance magnitude at 1 Hz; (**b**) phase at 2.5 kHz over the estrogen concentrations measured with sensor device with (I) (PEI/GO)_5_, (II) (PEI/mwcnt)_5_, and (III) (PAH⁄PAZO)_5_ film deposited on it at different estrogen concentrations.

**Figure 8 sensors-24-00481-f008:**
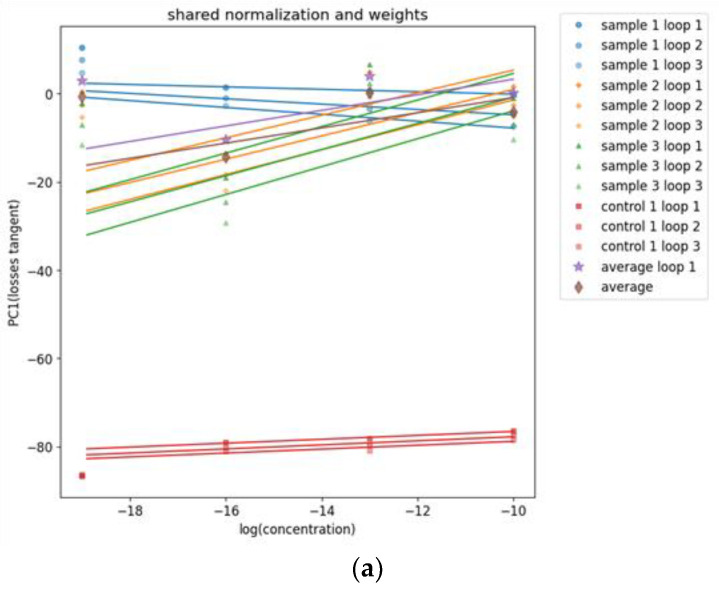
PC1 values over concentrations achieved with the best feature from the data measured with the sensor device covered with (**a**) (PEI/GO)_5_, (**b**) (PEI/mwcnt)_5_, and (**c**) (PAH⁄PAZO)_5_ thin films at different estrogen concentrations.

**Figure 9 sensors-24-00481-f009:**
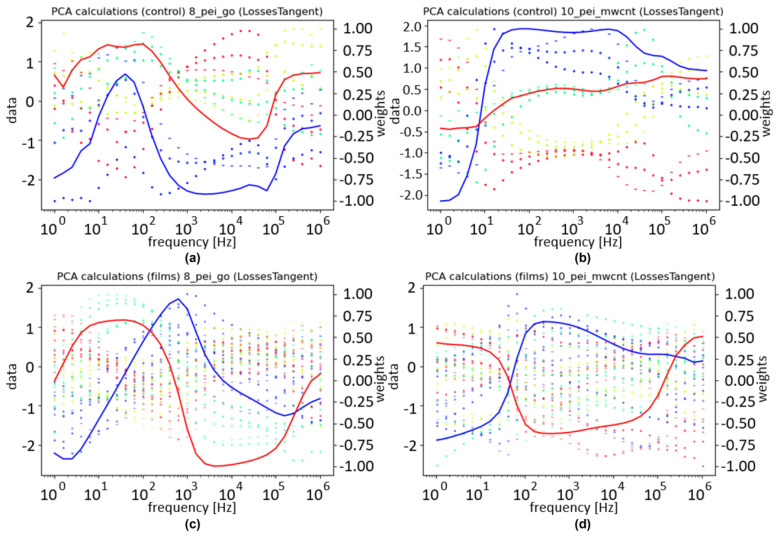
Normalized values and weights over frequency for the loss tangent with the sensor device covered with (**a**) (PEI/GO)_5_, (**b**) (PEI/mwcnt)_5_, and (**c**) (PAH⁄PAZO)_5_ thin films at different estrogen concentrations. The red and blue lines represent the weights for PC1 and PC2, respectively. The concentrations are differentiated by color, according to the legend.

**Figure 10 sensors-24-00481-f010:**
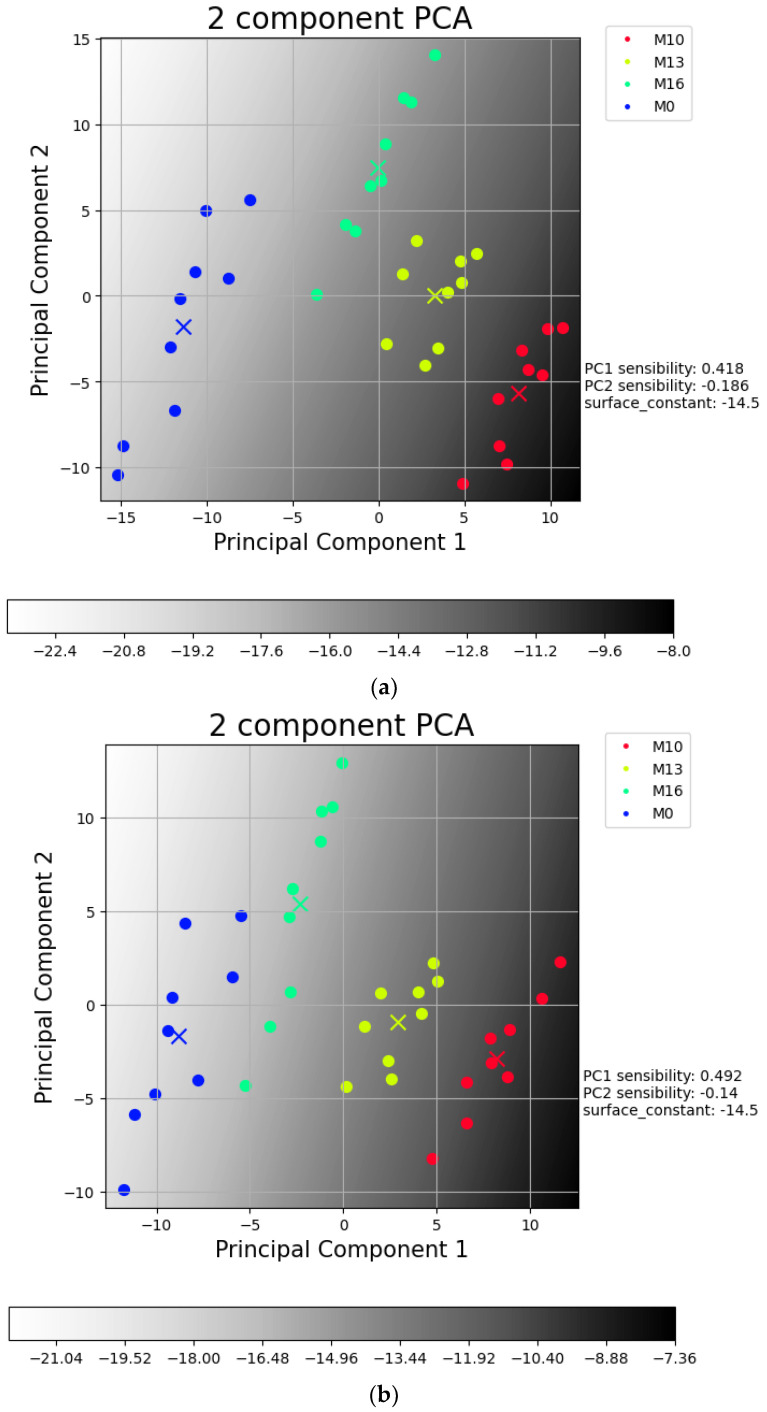
Plot with results achieved with sensor devices covered with (**a**) (PEI⁄mwcnt)_5_, (PEI⁄GO)_5_, and (PAH⁄PAZO)_5_; (**b**) (PEI⁄mwcnt)_5_ and (PEI⁄GO)_5_ thin films, using the loss tangent spectra. The x points are the mean values of each sample.

**Figure 11 sensors-24-00481-f011:**
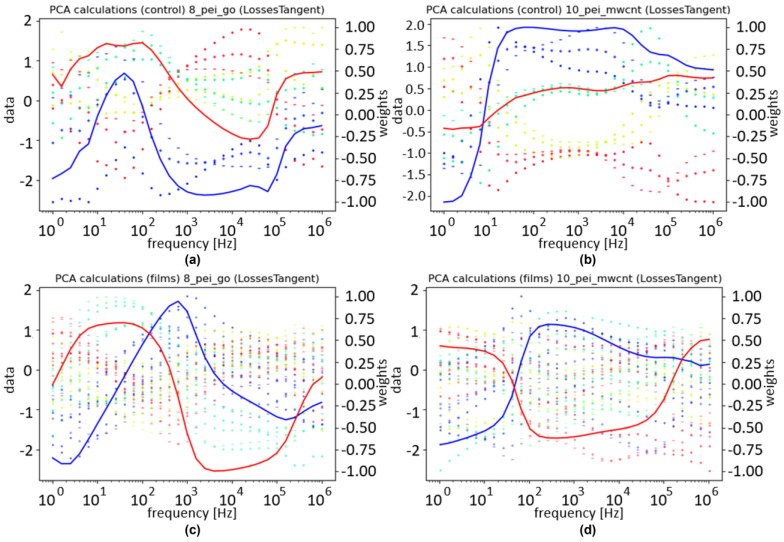
PCA data and weights: (**a**) PEI/mwcnt control; (**b**) PEI/mwcnt film; (**c**) PEI/GO control; (**d**) PEI/GO film. The red and blue lines represent the weights for PC1 and PC2, respectively. Blue, green, yellow, and red points represent the concentrations 0, 10^−16^, 10^−13^, and 10^−10^ mol/L, respectively. The color code of the samples is the same of the previous figures.

**Table 1 sensors-24-00481-t001:** Magnitude and phase at different estrogen concentrations for the PEI/mwcnt film at 1 Hz for the first loop.

Concentration (mol/L)	Control		Sample 1		Sample 2		Sample 3	
	Magnitude ×10^−5^ (Ω)	Phase (°)	Magnitude ×10^−5^ (Ω)	Phase (°)	Magnitude ×10^−5^ (Ω)	Phase (°)	Magnitude ×10^−5^ (Ω)	Phase (°)
0	7.19	−72.0	2.47	−83.0	2.79	−81.8	2.90	−83.0
10^−16^	7.85	−72.1	2.34	−83.8	2.54	−81.6	2.98	−81.6
10^−13^	8.06	−71.3	2.81	−83.0	2.63	−80.7	2.66	−82.0
10^−10^	10.54	−71.2	3.24	−82.2	2.68	−81.6	3.79	−80.0

## Data Availability

Data are contained within the article.

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
