# Peer review of "Graphene Oxide, Carbon Nanotubes, and Polyelectrolytes-Based Impedanciometric E-Tongue for Estrogen Detection in Complex Matrices"

_sensors, 2024, doi:10.3390/s24020481_

Round 1

Reviewer 1 Report

Comments and Suggestions for Authors

1. "e-tongue" system consists of sensor/s, data acquisition model, electronic circuit, computing elements etc. Authors should explain other components of e-tongue apart from sensors only as authors mentioned e-tongue in the paper title.

2. It is explain how authors are confirmed about the present of estrogen in tap water. Conventional analysis result to be included in the paper.

3. Different concentration of estrogen were presented in the paper. How authors calculated the concentration to be mentioned in the paper.

4. The table to be included in the paper indicating sensor output (may be only for best sensors) vs. different concentration of estrogen present in water.

5. Explanation/ experimentation need to be included in the paper about how the developed sensors are specific to estrogen and not detecting other chemical present in the water.

Comments on the Quality of English Language

Some sentences need to be rearrange for clear understanding such as line 28, 29, 34, 35, 111 etc.

Author Response

Response to the Comments

We thank the reviewers for the comments and suggestions that we think are properly addressed in the present form of the manuscript. We believe that corrections made fully accomplished reviewers comments and suggestions and improved its overall quality has been achieved.

Reviewer #1

  1. "e-tongue" system consists of sensor/s, data acquisition model, electronic circuit, computing elements etc. Authors should explain other components of e-tongue apart from sensors only as authors mentioned e-tongue in the paper title.

Answer: The paragraph now provides a further description of the e-tongue system.

  1. It is explain how authors are confirmed about the present of estrogen in tap water. Conventional analysis result to be included in the paper.

Answer:Thank you very much for your question, in the present study, we doped the tap water with different concentrations of estrogen molecules, and we did not perform conventional analysis of the water. If the used tap water presents an initial concentration of estrogen, the sensor will saturate at this concentration, and this was not observed in the present case. However, analysis of water revealed already the presence of estrogen. A new paragraph has been added that presents a study on the cost of Chile that verifies the presence of estrogen in water reserves.

  1. Different concentration of estrogen were presented in the paper. How authors calculated the concentration to be mentioned in the paper.

Answer: In section 2.2, a detailed explanation of the preparation of the estrogen solutions has been added to provide a better clarification about the estrogen solutions.

  1. The table to be included in the paper indicating sensor output (may be only for best sensors) vs. different concentration of estrogen present in water.

Answer: Table 1 has been added as per this suggestion.

  1. Explanation/ experimentation need to be included in the paper about how the developed sensors are specific to estrogen and not detecting other chemical present in the water.

Answer: Thank you very much for this question. Possibly some sensors can detect also other molecules as demonstrated in the article[https://doi.org/10.3390/s23010462], where it was demonstrated that PEI/GO thin films that allow the identification of acetic acid in the concentration range from 24 to 120 ppm, and of ethanol, isopropanol, and methanol in a concentration range from 18 to 90 ppm, respectively, and the quantification of acetic acid, ethanol, and isopropanol concentrations. Therefore, if a unique sensor allows the distinction of different molecules, the electronic/optics tongue and nose systems, in which an array of sensors is used, will give us a better distinction between different molecules and concentrations. This was better explained in the article.

Some sentences need to be rearrange for clear understanding such as line 28, 29, 34, 35, 111

Answer: The mentioned lines have been revised and corrected.

Reviewer 2 Report

Comments and Suggestions for Authors

The title indicates that the e-tongues employed only GO-based sensors.  However, this was not the case since other materials, such as carbon nanotubes and polyelectrolytes, were used.  Thus, the title has to be revised.  Furthermore, the transduction principle should also be included in the title.

The Introduction cited only the work done by their research group and did not include work done on impedance-based e-tongues by other authors.

The presentation of the results has to be improved to highlight the significant data.  The statistical validity of the inferences has to be supported by citing the results of significance tests. 

Comments on the Quality of English Language

The statements can be made more concise. In the present form, the presentation of the results is too detailed.

Author Response

Response to the Comments

We thank the reviewers for the comments and suggestions that we think are properly addressed in the present form of the manuscript. We believe that corrections made fully accomplished reviewers comments and suggestions and improved its overall quality has been achieved.

Reviewer #2

The title indicates that the e-tongues employed only GO-based sensors.  However, this was not the case since other materials, such as carbon nanotubes and polyelectrolytes, were used. Thus, the title has to be revised. Furthermore, the transduction principle should also be included in the title.

Answer: Thank you for your suggestion. The title has been changed to incorporate the proposed suggestions.

The Introduction cited only the work done by their research group and did not include work done on impedance-based e-tongues by other authors.

Answer: Thank you very much for your suggestion. References from the authors such as Taylor, D.M., Gomes, H.L., Riul, A., and Oliveira, O.N. have been added.

The presentation of the results has to be improved to highlight the significant data.  The statistical validity of the inferences has to be supported by citing the results of significance tests. 

Answer: References supporting the data treatment methods used have been added, namely

https://doi.org/10.1098/rsta.2015.0202, https://doi.org/10.1016/0169-7439(87)80084-9.

The statements can be made more concise. In the present form, the presentation of the results is too detailed.

Answer: The presentation of the results was improved to be more succinct.

Reviewer 3 Report

Comments and Suggestions for Authors

1. Figure 11 is not found in the manuscript. Please check the whole manuscript carefully.

2. SVM is mentioned in the manuscript only in the abstract, and SVM and its uses are not described in the rest of the manuscript.

3. Please describe in detail the use of the Damped Least Squares (DLS) method in the manuscript.

4. Please improve the clarity of the pictures in the manuscript.

5. The sensors in the electronic tongue are susceptible to environmental disturbances that can lead to drift, which reduces the detection performance. Therefore, in future research work, machine learning methods can be considered to design drift compensation methods to promote the intelligent process of sensors. Please refer to:

https://doi.org/10.1016/j.knosys.2022.110024

https://doi.org/10.1016/j.snb.2023.134716 

Author Response

Response to the Comments

We thank the reviewers for the comments and suggestions that we think are properly addressed in the present form of the manuscript. We believe that corrections made fully accomplished reviewers comments and suggestions and improved its overall quality has been achieved.

Reviewer #3

  1. Figure 11 is not found in the manuscript. Please check the whole manuscript carefully.

Answer: The numbering of the captions has been revised and corrected accordingly.

  1. SVM is mentioned in the manuscript only in the abstract, and SVM and its uses are not described in the rest of the manuscript.

Answer: Thank you for the detection of this mistake. SVM was never used in this work, the mentioned inference has been corrected and now reads as Damped Least Squares (DLS).

  1. Please describe in detail the use of the Damped Least Squares (DLS) method in the manuscript.

Answer: A small paragraph clarifying how the algorithm was used was added immediately after the paragraph that explains it.

  1. Please improve the clarity of the pictures in the manuscript.

Answer: The clarity of the figures in the manuscript was improved. Thank you.

  1. The sensors in the electronic tongue are susceptible to environmental disturbances that can lead to drift, which reduces the detection performance. Therefore, in future research work, machine learning methods can be considered to design drift compensation methods to promote the intelligent process of sensors. Please refer to:

https://doi.org/10.1016/j.knosys.2022.110024

https://doi.org/10.1016/j.snb.2023.134716 

Answer: Thank you very much for this information. The suggested references were very interesting and have been added along with a sentence to contextualize the proposal.

Reviewer 4 Report

Comments and Suggestions for Authors

            The article Graphene oxide-based e-tongue for estrogen detection in complex presents an evaluation of the use of advanced electrochemical sensors to detect trace amounts of estrogen in tap water. The Authors checked the effectiveness of sensors based on a wide range of polyelectrolytes, using impedance spectroscopy as a research tool. The results were interpreted using chemometric methods.

            I fully agree with the formal formulation of the research problem and the method of its solution, including the proposed selection of polyelectrolytes and the scope of empirical research.

            The issues raised by the Authors constitute one of the key challenges currently facing the global scientific community, focusing on topics related to environmental protection. For this reason, I would like to recommend the submitted manuscript for publication, with reference to the following questions and comments:

1. The choice of estrogen as a model xenobiotic is accurate. Is it possible to use the proposed solutions to identify other molecular systems (steroidal and non-steroidal)?

2. Although the selection of polyelectrolytes is accurate, the article lacks arguments for the selection of specific molecular systems.

3. What is the mechanism of interactions between estrogen molecules and the active sites of sensors? Wouldn't the answer to this question be enriched by proposing the results of appropriate theoretical calculations?

4. The names of the used substances (lines 65–67, 88–90) are incorrect (incorrect bracket hierarchy, unnecessary spaces and capital letters). What does the acronym "mwcnt" mean?

5. The reference to Figure 1 (line 91) is inaccurate and misleading.

6. Figure 1 A does not show the substance described. Figure 1 D refers to a compound that was not mentioned in the description (line 98), despite the lack of a disclaimer in the introductory text (lines 85–91).

7. Please suggest literature sources for paragraph 2.4 (lines 173–187, 188–201, 202–206).

8. Please remove the repeated text fragment (lines 257–271).

9. The language of the work contains a very large number of errors, mainly of an editorial nature: unjustified and inconsistent use of capital letters (lines 5, 6, 18, 20, 28, 29, 40, 46, 75, 115, 131, 133, 155, 161 and others), no indexes (line 63 and others) and no spaces (line 151 and others). There are also a few grammatical inconsistencies (lines 197, 204 and others).

            In my opinion, the article meets the high standards of the Journal. For this reason, I recommend the publication of the reviewed article in the Sensors after addressing the above questions and comments.

Comments on the Quality of English Language

The language of the work contains a very large number of errors, mainly of an editorial nature: unjustified and inconsistent use of capital letters (lines 5, 6, 18, 20, 28, 29, 40, 46, 75, 115, 131, 133, 155, 161 and others), no indexes (line 63 and others) and no spaces (line 151 and others). There are also a few grammatical inconsistencies (lines 197, 204 and others).

Author Response

Response to the Comments

We thank the reviewers for the comments and suggestions that we think are properly addressed in the present form of the manuscript. We believe that corrections made fully accomplished reviewers comments and suggestions and improved its overall quality has been achieved.

Reviewer #4

The article Graphene oxide-based e-tongue for estrogen detection in complex presents an evaluation of the use of advanced electrochemical sensors to detect trace amounts of estrogen in tap water. The Authors checked the effectiveness of sensors based on a wide range of polyelectrolytes, using impedance spectroscopy as a research tool. The results were interpreted using chemometric methods.

I fully agree with the formal formulation of the research problem and the method of its solution, including the proposed selection of polyelectrolytes and the scope of empirical research.

The issues raised by the Authors constitute one of the key challenges currently facing the global scientific community, focusing on topics related to environmental protection. For this reason, I would like to recommend the submitted manuscript for publication, with reference to the following questions and comments:

  1. The choice of estrogen as a model xenobiotic is accurate. Is it possible to use the proposed solutions to identify other molecular systems (steroidal and non-steroidal)?

Answer: Thank you very much for this question.  These films can detect different molecules and the proposed e-tongue with more sensors can distinguish other steroidal and non-steroidal molecular systems. Recently, in our group, we demonstrated that only PEI/GO thin films allow the identification of acetic acid in the concentration range from 24 to 120 ppm, and of ethanol, isopropanol, and methanol in a concentration range from 18 to 90 ppm, respectively, and the quantification of acetic acid, ethanol, and isopropanol concentrations. [https://doi.org/10.3390/s23010462], Therefore, if a unique sensor allows the distinction of different molecules, the electronic/optics tongue and nose concepts, in which an array of sensors is used, will give us a better distinction between different molecules and concentrations. This was better explained in the article.

  1. Although the selection of polyelectrolytes is accurate, the article lacks arguments for the selection of specific molecular systems.

Answer: Thank you very much for this question that is missing in the manuscript. These polyelectrolyte molecules were chosen since in our group we have already characterized these films. This information was included in the manuscript.

  1. What is the mechanism of interactions between estrogen molecules and the active sites of sensors? Wouldn't the answer to this question be enriched by proposing the results of appropriate theoretical calculations?

Answer: Thank you very much for this question. We did not concentrate in this work on the mechanisms of interaction between estrogen molecules and the active sites of the used sensors because this study needs other methods of analysis and possibly will be necessary to perform theoretical calculations.

  1. The names of the used substances (lines 65–67, 88–90) are incorrect (incorrect bracket hierarchy, unnecessary spaces and capital letters). What does the acronym "mwcnt" mean?

Answer: The new paragraph is more coherent with the literature, and now “mwcnt” was introduced as an acronym for multi-walled carbon nanotubes instead of carbon nanotubes.

  1. The reference to Figure 1 (line 91) is inaccurate and misleading.

Answer: the text that refers to the image has been corrected.

  1. Figure 1 A does not show the substance described. Figure 1 D refers to a compound that was not mentioned in the description (line 98), despite the lack of a disclaimer in the introductory text (lines 85–91).

Answer: The image has been corrected.

  1. Please suggest literature sources for paragraph 2.4 (lines 173–187, 188–201, 202–206).

Answer: The following references have been added in respect to the mentioned paragraph about the discretization error (doi.org/10.2514/6.2010-126). About DLS itself, it was explained that it is used through the Python function curve_fit from the scipy.optimize library.

  1. Please remove the repeated text fragment (lines 257–271).

Answer: The duplicated text has been removed from before Figure 4 and kept after.

  1. The language of the work contains a very large number of errors, mainly of an editorial nature: unjustified and inconsistent use of capital letters (lines 5, 6, 18, 20, 28, 29, 40, 46, 75, 115, 131, 133, 155, 161 and others), no indexes (line 63 and others) and no spaces (line 151 and others). There are also a few grammatical inconsistencies (lines 197, 204 and others).

Answer: All the lines mentioned have been revised and changed when appropriate to create a more coherent work.

In my opinion, the article meets the high standards of the Journal. For this reason, I recommend the publication of the reviewed article in the Sensors after addressing the above questions and comments.

The language of the work contains a very large number of errors, mainly of an editorial nature: unjustified and inconsistent use of capital letters (lines 5, 6, 18, 20, 28, 29, 40, 46, 75, 115, 131, 133, 155, 161 and others), no indexes (line 63 and others) and no spaces (line 151 and others). There are also a few grammatical inconsistencies (lines 197, 204 and others).

Answer: Thank you very much for your careful reading. The text has been revised and improved.

Round 2

Reviewer 1 Report

Comments and Suggestions for Authors

Real samples analysis is most important for development of any system.